# Modified CLEC3A-Derived Antimicrobial Peptides Lead to Enhanced Antimicrobial Activity against Drug-Resistant Bacteria

**DOI:** 10.3390/antibiotics12101532

**Published:** 2023-10-11

**Authors:** Denise Meinberger, Marco G. Drexelius, Joshua Grabeck, Gabriele Hermes, Annika Roth, Dzemal Elezagic, Ines Neundorf, Thomas Streichert, Andreas R. Klatt

**Affiliations:** 1Institute for Clinical Chemistry, Medical Faculty, University of Cologne, Kerpener Str. 62, 50937 Cologne, Germany; 2Institute for Biochemistry, Department of Chemistry, Faculty of Mathematics and Natural Sciences, University of Cologne, Zuelpicher Str. 47a, 50674 Cologne, Germany; 3Center for Molecular Biosciences, University of Cologne, Zuelpicher Str. 47a, 50674 Cologne, Germany

**Keywords:** CLEC3A, C-type lectin, antimicrobial peptides, peptide modification, drug-resistant bacteria, *MRSA*

## Abstract

Antimicrobial peptides (AMPs) represent a promising alternative to conventional antibiotics. Sequence changes can significantly improve the therapeutic properties of antimicrobial peptides. In our study, we apply different sequence modifications to enhance the performance of the CLEC3A-derived AMPs HT-16 and HT-47. We truncated their sequences, inserting a triple-glycine linker, adding an N-terminal tryptophan residue, and generating a D-amino acid variant, resulting in the generation of seven new peptides. We investigated their antimicrobial activity against gram-positive and gram-negative bacteria, their cytotoxicity to murine cells, and the biostability of the modified peptides in serum. We identified a novel antimicrobial peptide, WRK-30, with enhanced antimicrobial potency against *S. aureus* and *MRSA*. Additionally, WRK-30 was less cytotoxic to eukaryotic cells, allowing its application in higher concentrations in an in vivo setting. In conclusion, we identified a novel CLEC3A-derived antimicrobial peptide WRK-30 with significantly improved therapeutic properties and the potential to widen the repertoire of conventional antibiotics.

## 1. Introduction

The emergence of multi-drug resistant bacteria is a major threat to the healthcare system [1]. To counteract the impact of antimicrobial resistance, alternatives to common antibiotics, such as antimicrobial peptides (AMPs), are sorely needed [1,2]. AMPs are short peptides of 10 to 50 amino acids with, in the majority of cases, highly positive net charges [1]. Besides β sheet formations, loops, and extended random coils, AMPs often display amphipathic α helices as a secondary structure [3]. They exhibit potent antimicrobial activity and are commonly found as a component of immunity in various species [4], where they play an important role in pathogen defense. In mammals, they provide innate immunity by binding to the membranes of microbes, causing membrane disruption [5,6]. AMPs act by inducing membrane destabilization and pore formation, or by causing membrane lysis via the carpet model [5,6,7].

AMPs are most commonly synthesized as inactive precursors that become activated via proteolytic cleavage [8,9]. An example of these are peptides derived from C-type lectin domain family 3 member A (CLEC3A) [10], a cartilage-specific member of the C-type lectin superfamily [8]. Besides its carbohydrate recognition domain (CRD) encoded by exon 3, CLEC3A contains an α-helical oligomerization domain encoded by exon 2 as well as a 16 amino acid-long region with 8 positively charged amino acid residues and a signal peptide at the N-terminus encoded by exon 1 [10]. CLEC3A shows great similarity to AMP precursors and, in prior works, the CLEC3A-derived peptide HT-16, which includes the positively charged N-terminus, and HT-47, which additionally includes the α-helical oligomerization domain, have shown considerable antimicrobial activity against gram-positive and gram-negative bacteria [11]. The assessed antimicrobial activity of the peptides is comparable to the human cathelicidin LL-37, one of the best-studied AMPs [11]. However, in contrast to LL-37, HT-16 and HT-47 show no toxic effects to primary human cartilage cells after a two-hour incubation [11]. Moreover, coating the commonly used prosthetic material titanium with HT-16 and HT-47, respectively, significantly reduced the number of bacteria adhering to titanium [11].

In recent years, intensive research has been conducted on antimicrobial peptides and their therapeutic use. In addition to antimicrobial efficacy, cytotoxicity on eukaryotic cells plays a decisive role in their therapeutic application. LL-37, for example, plays a major role in the innate immune defense of bacterial infections. However, the cytotoxic properties of LL-37 on eukaryotic cells limit the use of this peptide for therapeutic applications. Another common disadvantage of natural AMPs is their low biostability as they are severely susceptible to proteases [12,13]. Sequence modifications can positively influence the therapeutic characteristics of antimicrobial peptides by enhancing their antimicrobial activity and biostability as well as reducing their cytotoxicity.

With the aim to identify novel AMPs with improved therapeutic properties, here we apply different sequence modifications of the CLEC3A-derived AMPs HT 16 and HT 47 to enhance their antimicrobial activity against *P. aeruginosa*, *S. aureus*, and in particular, against *methicillin-resistant S. aureus* (*MRSA*). Furthermore, we investigate the cytotoxicity and biostability of the modified CLEC3A-derived peptides as these are important aspects of in vivo applications.

## 2. Results

### 2.1. Modifying CLEC3A-Derived Antimicrobial Peptides

To enhance the antimicrobial efficacy of the CLEC3A-derived AMPs HT-16 and HT-47 we applied different strategies resulting in the generation of a total of six different novel peptides (Figure 1, Table 1). First, we used truncations, since Elezagic et. al. (2019) [11] showed that the positively charged N-terminus of CLEC3A encoded by exon 1, which is contained in HT-16 and HT-47, is essential for the antimicrobial activity of CLEC3A-derived peptides. Therefore, we eliminated parts of the N-terminus that showed only a low positive charge density, leading to the formation of the peptide RK-9 (Figure 1, Table 1). For HT-47, we also truncated the α-helical part encoded by exon 2. We used the prediction tool Network Protein Sequence Analysis from prabi [14] and visually assessed the amphipathicity with Helical Wheel Projections from NetWheels (http://lbqp.unb.br/NetWheels, accessed on 1 April 2021) (Appendix A), leading to the formation of the peptide RK-31 (Figure 1, Table 1). HT-47 kills bacteria by permeabilizing the bacterial membrane, which finally leads to leakage of cell contents [11]. Thus, HT-47 binds to LPS or LTA on the bacterial membrane by its positively charged N-terminus followed by the insertion of its amphipathic α-helix into the membrane [11]. However, this mechanism requires a certain flexibility between the N-terminal positively charged domain and the α-helix. Since sequence shortening can cause steric tension, peptides might lose their flexibility and activity. Therefore, a small, flexible triple-glycine-linker between the positively charged region and the α-helix was introduced, resulting in the formation of the peptide RK-30 (Figure 1, Table 1). As carrying a free tryptophan residue was reported to increase hydrophobicity and facilitate incorporation of the peptide into the bacterial membrane, thereby improving AMP specificity [15,16], we further added a tryptophan residue to the N-termini of RK-9, RK-31, and RK-30, resulting in the formation of the new peptides WRK-9, WRK-31, and WRK-30 (Figure 1, Table 1). To assess the modified peptides, we used LL-37, which has already entered clinical trials, as a positive control as it has a similar length, and its antimicrobial activity is promoted via a positive charge and an amphipathic α-helix. As negative controls, ST-16 (Ex1z), which corresponds to HT-16 with abolished positive charge, and DK-29 (Ex2), which is solely the amphipathic α-helix of HT-47 without the positive charge of the N-terminus, were used [11].

### 2.2. Assessing the Antimicrobial Activity

We assessed the antimicrobial activity of the modified CLEC3A-derived peptides using an antimicrobial activity assay and determined their MIC50 (Figure 2). We used the well-investigated AMP LL-37 as a positive control. As negative controls, we used ST-16 (Ex1z), which corresponds to HT-16 with an abolished positive charge, and DK-29 (Ex2), which is solely the amphipathic α-helix of HT-47 without the positive charge of the N-terminus. Neither ST-16 nor DK-29 display any antimicrobial activity [11]. First, we investigated the antimicrobial activity of the peptides modified from HT-16: RK-9 and WRK-9 (Figure 2a–c). Both peptides exhibited lower antimicrobial activity than the native HT-16 against *P. aeruginosa*, *S. aureus*, and *methicillin-resistant S. aureus* (*MRSA*) with significantly higher MIC50 concentrations.

Moreover, we tested the antimicrobial activity of peptides modified from HT-47: RK-31, WRK-31, RK-30, and WRK-30 (Figure 3a–c). The truncated peptides RK-31 and WRK-31 did not inhibit the growth of *P. aeruginosa* and *MRSA* and only killed *S. aureus* in much higher concentrations than the native peptides. On the other hand, the truncated peptides with a triple glycine sequence, RK-30, and WRK-30, exhibited a more potent antimicrobial activity. Namely, RK-30 and WRK-30 showed strong antimicrobial activity against *P. aeruginosa* and *S. aureus*, with a MIC50 comparable to that of HT-47. Remarkably, both peptides showed a stronger antimicrobial activity against *MRSA* than the native HT-47, with MIC50 decreasing by 2.4-fold (RK-30) and even 3.7-fold (WRK-30). Adding a single tryptophan residue to the N-terminus enhances the antimicrobial activity of a peptide and reduces the MIC50 by at least 30%.

We successfully modified the peptide HT-47 to WRK-30, which exhibited a comparable antimicrobial activity against gram-negative *P. aeruginosa* and a strongly enhanced activity against gram-positive *S. aureus* and *MRSA*.

### 2.3. Interaction of WRK-30 with Giant Unilamellar Vesicles

To elucidate the mechanism of action of WRK-30, we performed lipid–peptide interaction experiments using fluorescence microscopy with giant unilamellar vesicles (GUVs) composed of anionic lipids mimicking bacterial membranes. Without incubation with the peptide, we produced intact GUVs, which exhibited no autofluorescence in the green channel, used to visualize the CF-labeled WRK-30 (Figure 4). After a 30 min incubation with 1 µM WRK 30, GUVs were still intact as no Oyster405 leaked out of the GUVs and the membrane stained with Atto550 did not show any sign of deformation (Figure 4). However, CF-labeled WRK 30 could be found accumulated on the membrane, showing binding of WRK-30 to the membrane and therefore membrane activity. Similarly, no leakage of vesicle content could be found after incubation with 5 µM of WRK-30, however the membrane showed early signs of lysis (Figure 4). Incubating the GUVs with 10 µM of WRK-30 led to complete lysis of the GUVs as we could not find any remaining vesicles but only lipid debris.

### 2.4. Determining the Cytotoxicity

In order to assess the suitability of the peptides for non-systemic in vivo applications in mice, we determined the cytotoxicity of the modified CLEC3A-derived peptides against murine fibroblasts (NIH3T3 cells) (Figure 5). As the modified CLEC3A-derived peptides RK-31 and WRK-31 were not as effective as antimicrobials as the other novel variants, we did not include them in these experiments. First, we tested the modified CLEC3A-derived AMPs in concentrations corresponding to the MIC50 of HT-47 against *S. aureus*, as well as 10-fold and 100-fold of the MIC50 of HT-47 against *S. aureus*. The three concentrations (1-fold, 10-fold, 100-fold MIC50) are the concentrations that will be tested in future in vivo studies. Incubation of NIH3T3 cells with the modified CLEC3A-derived peptides did not impair cell survival or growth (Figure 5A). However, HT-47 showed a concentration-dependent tendency for cytotoxicity (Figure 4). Additionally, we could confirm the cytotoxic effects of LL-37 at a concentration of 30 µM (Figure 5A). Except for HT-47, incubating the murine fibroblasts for a longer time (96 h) with the peptides did also not lead to any cytotoxicity (Figure 5B). Finally, we used even higher concentrations of 40 µM, 60 µM, and 80 µM to investigate maximal non-cytotoxic concentrations. Although HT-47 showed a concentration-dependent cytotoxic tendency, it did not prove to be statistically significant. On the other hand, a concentration of up to 80 µM of WRK-30 did not show any cytotoxic effects (Figure 5C).

### 2.5. Unraveling the Secondary Structure

The secondary structure of the peptides can reveal potential structure–activity relationships. Therefore, we performed circular dichroism spectroscopy in an aqueous phosphate buffer (PB) and a more hydrophobic environment in a buffer containing 50% trifluoroethanol (TFE) (50% TFE in PB). In phosphate buffer, as well as in 50% TFE in PB, the short peptides HT-16, RK-9, and WRK-9 showed no structuration (Appendix A). Therefore, even under more hydrophobic conditions, these peptides do not form an α-helical structure.

We have also investigated the secondary structure of the longer peptides by CD analysis. The low intensities indicate a small population of peptides with a helical structure. We used the R-value and fractional helicity (FH) to assess the formed structures of the peptides. HT-47 displays an α-helical formation with an R-value of 0.485 and a fractional helicity (FH) of 17.9% in PB (Figure 6a). R-values of 1 describe a perfectly formed α-helix. The formed α-helix of HT-47 stabilized under more hydrophobic conditions, as indicated by the increased R-value of 0.768 and the increased FH of 37.0% in 50% TFE in PB (Figure 6b). In contrast, RK-31, WRK-31, and RK-30 were unstructured in PB but adopted an α-helical structure in a more hydrophobic environment (Appendix A) with an R-value under 0.361 and FH up to 8.68%. Comparable to RK-31, WRK-31, and RK-30, WRK-30 showed no structuration at all when in buffer solution only, which is underlined by the R-value of 0.239 and FH of 1.0% (Figure 6a). However, in a more hydrophobic environment, the secondary structure of WRK-30 converted to an α-helix, even though the effect was weaker than for HT-47 (R-value of 0.710 and FH of only 8.7%) (Figure 6b). Comparison of CD analysis showed a greater number of helical peptides and a more perfect formation of an α helix, and thus the formation of a more stable α helix of HT-47 compared with the modified CLEC3A-based peptides.

### 2.6. Investigating the Biostability

We investigated the biostability of WRK-30 in murine serum, as AMP stability in physiological media is another important aspect for in vivo applications (Figure 6c,d). Therefore, we incubated WRK-30 and HT-47 for several time periods in murine serum and detected the remaining peptide via immunoblot analysis. It is important to point out that HT-47 and WRK-30 migrate differently in buffer compared to serum in SDS-PAGE due to the different protein load (Figure 6, Figure 7 and Appendix A). The native peptide HT-47 was detectable for up to 8 h in murine serum (Figure 6c) but disappeared after 24 h (not shown). In contrast, WRK-30 was only detectable for up to 30 min (Figure 6d). The rapid degradation of WRK-30 is likely due to its enhanced susceptibility to serum proteases [17,18] and its less stable α-helical structure.

### 2.7. Enhancing the Biostability—The Second Round of Peptide Modification

To improve the biostability of WRK-30, we generated the D-amino acid form of WRK-30, namely dWRK-30, since non-proteinogenic D-variants have been shown to reduce the susceptibility of peptides to proteases [19]. First, we tested the antimicrobial activity of dWRK-30 and observed that substituting all L-amino acids with D-amino acids led to comparable antimicrobial activity as for WRK-30 (Figure 8a–c).

We then investigated the cytotoxicity of dWRK-30, which, contrary to WRK-30, exhibits a cytotoxic effect when applied in high concentrations (Figure 7a). This effect is unlikely due to a difference in their secondary structure since the recorded CD-spectrum of dWRK-30 is identical to that of WRK-30 except for the inversion, caused by the D-amino acids (Figure 7b). Even the R-value of 0.256 and FH of −1.5% in PB and the R-value of 0.502 and FH of −3.7% in 50% TFE in PB of dWRK-30 (Figure 7b) are similar to the values measured for WRK-30 (Figure 6a,b). In terms of biostability, however, dWRK-30 shows a clear advantage in comparison to WRK-30. Generating a D-amino acid form leads to a prolonged biostability of dWRK-30 in murine serum of up to 96 h (Figure 7c), compared to that of 30 min of WRK-30 (Figure 6d).

## 3. Discussion

The number of newly approved antibiotics has decreased significantly in the last few decades and scientists are warning of a “post-antibiotic age”. AMPs are promising candidates for the treatment of bacterial infections [20] and several antimicrobial peptides such as LL-37, nisin, or melittin are currently being investigated in clinical trials [21]. In addition to their antimicrobial activity, their biostability and cytotoxicity for eukaryotic cells play an important role in potential therapeutic applications. Furthermore, the length of the peptides plays a crucial role in production costs and thus profitability. In this study, we modified the CLEC3A-derived AMPs HT 16 and HT 47 and assessed their therapeutic characteristics.

AMPs have core sequences that are critical for their antimicrobial activity. A crucial characteristic of AMPs is their positive net charge and a high number of consecutive lysins and arginines [22]. We have therefore truncated HT-16 by the first 7 amino acids, which include only one lysine and one arginine. The truncation of the first 7 amino acid residues of HT-16 resulted in the formation of peptides with reduced antimicrobial activity, indicating a loss of amino acids important for antimicrobial activity. By CD analysis, we could show that HT-16, RK-9, and WRK-9 are entirely unstructured and do not form an α helix. Thus, their mechanism of action cannot rely on amphipathic α helices, but possibly on a carpet model, by which the peptides accumulate at the membrane surface and disrupt the lipid layers, instead of permeabilizing it [22]. The net charge of HT-16 is +8, while the net charge of RK-9 and WRK-9 is +6. We may therefore have excluded important positively charged amino acids that are necessary for the interaction with the bacterial cell membrane. However, it is likely that uncharged amino acids also contribute to an interaction with the bacterial membrane, for example as spacers, in order to organize the positively charged amino acid residues in the optimal position to bind the negatively charged lipids of the bacterial wall.

Another important structural feature of AMPs is the formation of an α-helix. In the case of HT-47, exon 2 codes for the α-helical sequence consisting of 31 amino acids. Analysis of exon 2 with a prediction tool revealed a core sequence of 18 amino acid residues critical for the formation of an α-helix. Therefore, we truncated the α-helical part of HT-47 down to the core sequence of the 18 amino acid residues. CD analyses showed that HT-47 forms an α-helix in phosphate buffer and TFE buffer. In contrast, WRK-30 with a truncated α-helix is unstructured in phosphate buffer but possesses an α-helical structure in a more hydrophobic environment. Since WRK-30 exhibits enhanced antimicrobial activity compared to HT-47, we conclude that the formation of an α-helix in a hydrophobic environment is more important for antimicrobial activity than a permanent α-helix and that the α-helix of WRK-30 forms after binding to the bacterial wall.

The molecular mechanism of action of the native CLEC3A-derived peptide HT-47 includes binding of the bacterial membrane through the positively charged region of the peptide and an insertion of the amphipathic α-helix into the bacterial membrane, leading to cell lysis via pore formation [11]. For the α-helix to be inserted into the membrane, there needs to be enough flexibility between it and the positively charged domain. To improve the flexibility between the positively charged N-terminal domain and the α-helical C-terminal part, we introduced a triple-glycine linker instead of the native linker with the amino acid sequence DKDGD. Although a double-glycine linker would have had a similar beneficial effect as a triple glycine linker, the addition of more glycine residues would probably have the effect of a hyperflexible linker that does not allow targeted insertion of the α helix into the membrane. Truncation of the α-helix combined with the native DKDGD-linker probably led to peptide stiffness and therefore decreased antimicrobial activity of the modified peptides RK-31 and WRK-31. The insertion of a triple-glycine linker seems to solve the issue of steric tension as the modified peptides RK-30 and WRK-30 showed enhanced antimicrobial activity. Moreover, the native linker, which consists of the amino acid sequence DKDGD, has three negatively charged amino acid residues (D), in contrast to the triple-glycine linker. It is therefore conceivable that the three negatively charged amino acid residues of the DKDGD-linker lead to the electrostatic repulsion of negatively charged molecules on the bacterial membrane. The addition of a triple-glycine linker to the peptides, therefore, is likely to lead to an enhanced and more stable membrane binding.

Tryptophan, a highly hydrophobic amino acid, plays an important role in strengthening the binding and perturbation of AMPs to the bacterial membrane [5,23,24]. Although the C-terminal part of our peptides, namely the α-helix, is in charge of membrane insertion, a C-terminal placement of tryptophan in AMPs with amphipathic α-helices has been reported to not influence the antimicrobial activity [25]. On the other hand, the addition of an N-terminal tryptophan residue to cationic AMPs enhances their antimicrobial activity [16]. The indole ring found in tryptophan has the ability to interact with the headgroups of phospholipids of membranes. Additionally, due to the quadrupole of the indole ring, tryptophan residues tend to favor the interfacial region of lipid bilayers. This allows for interactions between the π–electron system of tryptophan and neighboring cations such as arginine, resulting in a cation–π interaction. This interaction leads to arginine being more firmly anchored within the lipid bilayer, resulting in a prolonged association with the membrane [16,26]. The addition of a tryptophan residue to CLEC3A AMPs indeed resulted in an enhanced antimicrobial activity with a reduction of MIC50 values of at least 30%.

The modified peptide WRK-30 showed enhanced antimicrobial activity but decreased biostability compared to the native HT-47 in murine serum. Since proteolytic enzymes usually catalyze stereospecific reactions, the use of D-amino acids may improve biostability [27]. Therefore, we substituted all amino acids of WRK-30 with D-amino acids (dWRK-30). Indeed, dWRK-30 was more stable than the L-amino acid version WRK-30. However, this led to a significant increase in cytotoxicity of dWRK-30 compared to WRK-30. The longer biostability is possibly associated with increased cytotoxicity, as cells are exposed to the cytotoxic effect of the peptide for a longer time. Therefore, it is important to always assess the therapeutic properties of AMPs, such as antimicrobial activity, biostability, and cytotoxicity as a whole, rather than separately. The possibility of using dWRK-30 in in vivo applications at a lower concentration but for longer periods due to its longer biostability does not seem reasonable, as lower concentrations do not exhibit sufficient antimicrobial activity.

For potential in vivo applications, we investigated the cytotoxicity of the peptides toward eukaryotic cells. All of the modified CLEC3A-derived peptides showed less cytotoxicity towards murine fibroblasts compared to the native CLEC3A-derived peptides. Even in high concentrations up to 80 µM, WRK-30 was not cytotoxic, in contrast to the native HT-47. The positively charged HT-16 does not show any signs of cytotoxicity. Our results indicate that the interaction of amphipathic α-helices with membranes is the cause of the cytotoxic tendency of HT-47. Furthermore, in light of our CD spectroscopy results, the less stable amphipathic α-helix of WRK-30 compared to that of HT-47 could be the reason for the decrease in cytotoxicity of WRK-30. A more stable amphipathic α-helix might incorporate itself better into the membrane of eukaryotic cells. Additionally, a pre-formed α-helix in a hydrophilic solution, as in the case of HT-47, might facilitate an even faster transition to a fully formed α-helix in the more hydrophobic environment of a eukaryotic cell membrane.

Modifications to HT-47 resulted in WRK-30 with enhanced antimicrobial activity and decreased cytotoxicity. However, this led to a decreased biostability of WRK-30 in murine serum compared to HT-47. Low biostability and increased degradation levels are common challenges in the in vivo use of antimicrobial peptides. Still, it is important to note that antibiotics commonly used in clinical settings only have a serum half-life of 1–2 h [28]. To overcome these challenges, additional adjustments to WRK-30, such as creating derivatives with non-natural side chains, incorporating β-amino acids, or modifying the N- and C-termini, could improve the peptide’s stability in serum [29,30,31]. Furthermore, it should be noted that AMPs are well-suited for local applications, where different biostabilities may be observed as compared to serum.

In conclusion, shortening the α-helix of CLEC3A-derived HT-47, introducing a triple-glycine linker between the N-terminal domain and the C-terminal α-helix, and adding an N-terminal tryptophan residue to the peptide sequence resulted in the formation of the novel AMP WRK-30 with improved therapeutic properties. WRK-30 shows significantly improved antimicrobial activity, in particular against *methicillin-resistant S. aureus*, as well as reduced cytotoxicity, making it a suitable candidate for future in vivo experiments.

## 4. Materials and Methods

### 4.1. Bacterial Strains

*Pseudomonas aeruginosa* (Psae-27853), *Staphylococcus aureus* (ATCC-29213), and *methicillin-resistant Staphylococcus aureus* (MRSA-43300) were used. Bacteria were cultivated in tryptic soy broth (TSB) (Merck, Darmstadt, Germany) at 37 °C and 225rpm or on TSB–agarose plates at 37 °C.

### 4.2. Peptide Design/Modification

The amino acid sequence of CLEC3A-derived peptides was modified in line with their secondary structure using Network Protein Sequence Analysis tool from prabi [14] and Helical Wheel Projections from NetWheels “http://lbqp.unb.br/NetWheels (accessed on 1 April 2021)”. First, CLEC3A-derived peptides were truncated. Second, a small, flexible glycine-linker was introduced, instead of the endogenous linker, to modified peptides of the CLEC3A-derived HT-47. Third, a tryptophan residue was added to the N-terminus. Lastly, D-amino acids were introduced to the modified peptide WRK-30.

### 4.3. Peptide Synthesis

Peptides were synthesized on Fmoc-Wang resin beads to receive peptides with a free C-terminus. The first amino acid was already coupled to the resin with a loading of 0.4–0.8 mmol/g. Further, amino acids were coupled using an automated peptide synthesizer (MultiSynTech I, Biotage) by double coupling steps with 8 equivalents (eq.) Fmoc-aa-OH, Oxyma pure^®^, and N,N′-diisopropylcarbodiimide (DIC). Aspartic acid and glycine (DG)-motives were coupled manually as a Fmoc-Asp(OtBu)-(Dmb)Gly-OH dipeptide using hexafluorophosphate azabenzotriazole tetramethyl uronium (HATU) (2 eq.) and N,N-diisopropylethylamine (DIPEA) (2 eq.) in N,N-dimethylformamide (DMF). Then, peptide synthesis was finalized using the automated peptide synthesizer. Peptides HT-16, RK-9, and WRK-9 were removed from the resin using trifluoroacetic acid (TFA)/triisopropylsilane (TIS)/H_2_O (95:2.5:2.5 *v*/*v*/*v*) for 3 h, followed by precipitation in ice-cold diethyl ether. The other peptides were removed from the resin using TFA/ethanedithiol/thioanisole (90:3:7) and then precipitated in ice-cold diethyl ether. The peptides were purified by reversed-phase high-performance liquid chromatography (RP–HPLC) with a C18 column using acetonitrile in water (0.1% TFA), and fractions were analyzed by analytical liquid chromatography–mass spectrometry (LC–MS). The final purity of all peptides was 99%, except for that of HT-47, with 91% (Appendix A).

Peptides LL-37, ST-16 (Ctrl 1), HT-47, and dWRK-30 were custom ordered from Genosphere (Paris, France) with a purity of >95%. Peptide DK-29 (Ctrl 2) was custom-ordered from Biomatik (Wilmington, NC, USA) with a purity of >95%.

### 4.4. Antimicrobial Activity Assay

Bacteria were grown to an OD_620_ of 0.5, harvested by centrifugation, and washed with tris-glucose buffer (TG buffer: 10 mM tris, 5 mM glucose; pH = 7.4) (*P. aeruginosa* and *S. aureus*) or phosphate–glucose buffer (PG buffer: 10 mM K_2_HPO_4_, 5 mM glucose; pH = 7.4) (*MRSA*). Bacteria were adjusted to 2 *×* 10^6^ colony forming units (CFU)/mL and were incubated in 1:2 dilutions for 2 h at 37 °C with CLEC3A-derived and modified CLEC3A-derived peptides in TG buffer or PG buffer. The known AMP LL-37 was used as a positive control, while the CLEC3A-derived peptides ST-16 (Ctrl 1) and DK-29 (Ctrl 2) and untreated bacteria were used as negative controls. The bacteria were then plated onto TSB–agarose plates in previously determined dilutions and cultured overnight at 37 °C. The number of grown colonies was determined with the ImageJ cell counter plugin and is presented as a percentage of the number of grown colonies from the untreated control. The minimal inhibitory concentration (MIC50) was defined as the concentration of antimicrobial peptide where only 50% of bacteria survived. Used peptide concentrations are provided in Appendix A.

### 4.5. Interaction of WRK-30 with Giant Unilamellar Vesicles

Giant unilamellar vesicles (GUVs) were produced by coating a clean microscope slide with 200 µL of 1% super low melting agarose and drying the slide for 30 min at 50 °C. 10 mL of a lipid mixture containing 30 mol% DOPG, 40 mol% DOPC, 30 mol% DOPE (Avanti Polar Lipids, Inc. Alabaster, AL, USA), and 0.2 mol% Atto550 labeled DOPE (Atto Tec, Siegen, Germany) in chloroform was added onto the thin agarose layer and dried for 1 h using an exicator under vacuum. A sealing ring was placed around the lipid film and the lipids were hydrated for 2 h at room temperature with 300 µL dextran buffer (10 mM HEPES buffer (pH 7.4), 50 mM KCl, 50 mM NaCl and 1 mg/mL Dextran) containing 3 µL Oyster405 (Luminaris GmbH, Münster, Germany). Afterward, the hydrated lipid solution was transferred into 1.5 mL tubes and centrifuged for 10 min at maximum speed. The supernatant was discarded and the pellet was resuspended in 300 µL dextran buffer. For fluorescence microscopy, 40 µL of GUV solution mixed with 0 µM, 1 µM, 5 µM, and 10 µM 5,6-carboxyfluorescein (CF)-labeled WRK-30 filled up to 100 µL with dextran buffer were added to an eight well Ibidi^®^ plate. After a 30 min incubation with the peptide at room temperature 100 µL of dextran buffer was added to each well and pictures of the GUVs were taken using a BZ-X800E microscope (Keyence, Osaka, Japan). Pictures were processed with ImageJ to subtract the background.

### 4.6. Cytotoxicity Assay

NIH3T3 cells (murine fibroblasts), obtained from Prof. Dr. Wielckens, were seeded and precultured overnight. For 24 h incubation with peptides, a density of 1 *×* 10^4^ cells/well and for 96 h incubation with peptides a density of 5 *×* 10^3^ cells/well was used. After preculturing, the culture medium was replaced and cells were incubated with a peptide-containing medium for 24 h and 96 h. Afterward, the peptide-containing medium was removed and cells were washed with PBS before adding CellTiter 96^®^ AQueous One Solution (Promega, Madison, WI, USA) containing 3-(4,5-dimethylthiazol-2-yl)-5-(3-carboxymethoxyphenyl)-2-(4-sulfophenyl)-2H-tetrazolium (MTS) to the culture medium. Cells incubated with peptides for 24 h were incubated with MTS medium for 1 h at 37 °C, and absorbance was measured at 490 nm. Cells incubated with peptides for 96 h were incubated with MTS medium for 2 h at 37 °C before measuring the absorbance. Samples were measured in triplicates, and the cell viability of three individual experiments is presented as a percentage of the measured OD from the untreated control. As further controls, vancomycin (final concentration 7.35 µg/mL), colistin (final concentration 147 µg/mL), and 1% TritonX in culture medium were used. Used peptide concentrations are provided in Appendix A.

### 4.7. Circular Dichroism Spectroscopy

Peptides were diluted in 10 mM phosphate buffer (PB) (pH 7.4) or PB (pH 7) containing 50% trifluoroethanol (50% TFE in PB) to a final concentration of 20 µM and 100 µM. Samples were measured in a 1 mm thick quartz cuvette and spectra were recorded from 180 to 260 nm. The molar ellipticity was calculated from θ_measured_ in degree using the following equation:[θ] = θ_measured_ × M_peptide molecular weight_/(10 × c_peptide_ × d_cuvette_) = degrees (deg)∙cm^2^∙dmol^−1^

R-values were calculated using the equation R = θ_222 nm_/θ_207 nm_ to determine the quality of a formed α-helix. An R-value of 1 represents a perfect α-helix.

The fractional helicity (FH) of the peptides was calculated via the mean residue ellipticity at 222 nm using the equation FH = (θ_222 nm_ − θ_u_)/(θ_h_ − θ_u_), wherein θ_u_ = −3000 represents the value of θ at 222 nm if the measured peptide is 0% helical and θ_h_ = −39,500 represents the value of θ at 222 nm if the measured peptide is 100% helical [32].

### 4.8. Biostability Assay

The peptides (HT-47, WRK-30, and dWRK-30) were mixed with pooled murine serum, obtained from C57BL6/N wildtype mice via cardiac puncture immediately after death by cervical dislocation (the procedure was approved by the local government authority LANUV under permit no. UniKöln_Anzeige§4.22.001), to a final concentration of 30 µM. In parallel, a control with water and pooled murine serum was prepared. Both mixtures were incubated at room temperature, and samples were taken at the time points indicated in Appendix A. The samples were immediately added to a Novex Tricine SDS Sample Buffer containing a NuPage Sample Reducing Agent and incubated at 85 °C for 2 min. Boiled samples were stored at 4 °C. After the collection of all time points, samples were separated into 10–20% Tricine Gels. Gels with HT-47 and WRK-30 were subsequently transferred onto PVDF membranes (0.2 µm pore size) and immunoblot analysis with 1:200 dilutions of a custom-ordered and affinity-purified rabbit anti-WRK-30 (Eurogentec, Seraing, Belgium) antibody was performed. Gels with dWRK-30 were stained with Coomassie brilliant blue.

### 4.9. Statistical Analysis

Statistical analysis of results was performed with Prism 9.3.1 (471) (GraphPad, San Diego, CA, USA). All tables and graphs show calculated averages and standard deviations obtained from three individual experiments. Bar charts of the MIC50 include numerical values with confidence intervals (95%) as the legend. Statistical significance was determined with a paired ANOVA test followed by a Dunnett test and multiple comparisons, comparing each peptide with the respective parent peptide HT-16, HT-47, or WRK-30 as indicated in the figure captions.

## Figures and Tables

**Figure 1 antibiotics-12-01532-f001:**
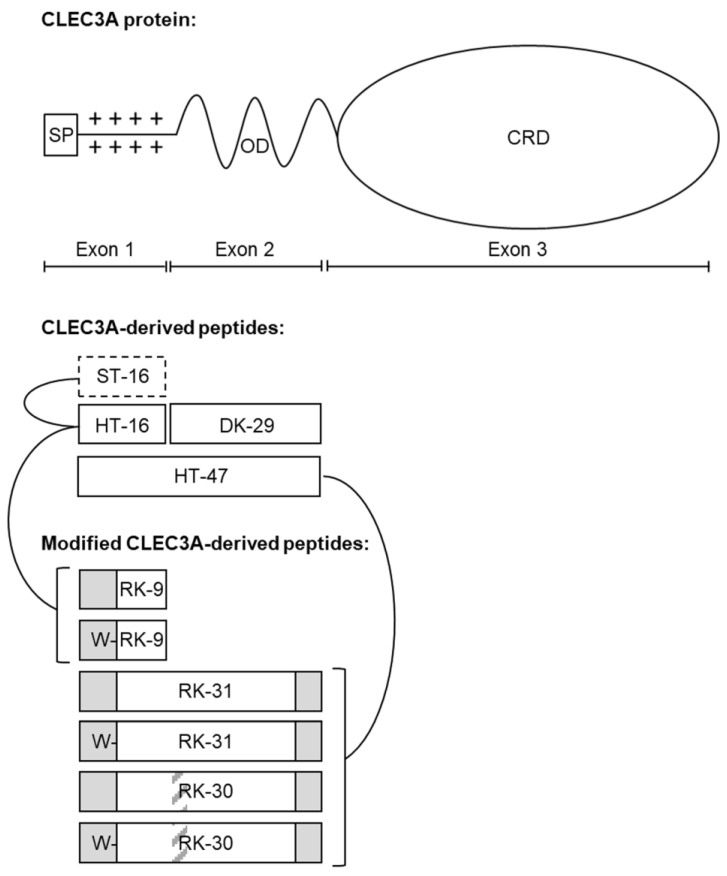
Modified CLEC3A-derived peptides. Whole CLEC3A protein with signal peptide (SP), oligomerization domain (OD), and carbohydrate recognition domain (CRD). CLEC3A-derived peptides (ST-16, HT-16, DK-29, and HT-47) aligned to their origin in the CLEC3A protein. Modified CLEC3A-derived peptides (RK-9, WRK-9, RK-31, WRK-31, RK-30, and WRK-30) are depicted underneath, with lines linking them to their original peptide. Dashed black framed boxes indicate abolishment of positive net charge, grey regions in black framed boxes indicate truncations, grey striped areas indicate inserted linkers, and Ws in front of the white regions in black-framed boxes indicate added tryptophan residues. Adapted from Elezagic et al. (2019) [11].

**Figure 2 antibiotics-12-01532-f002:**
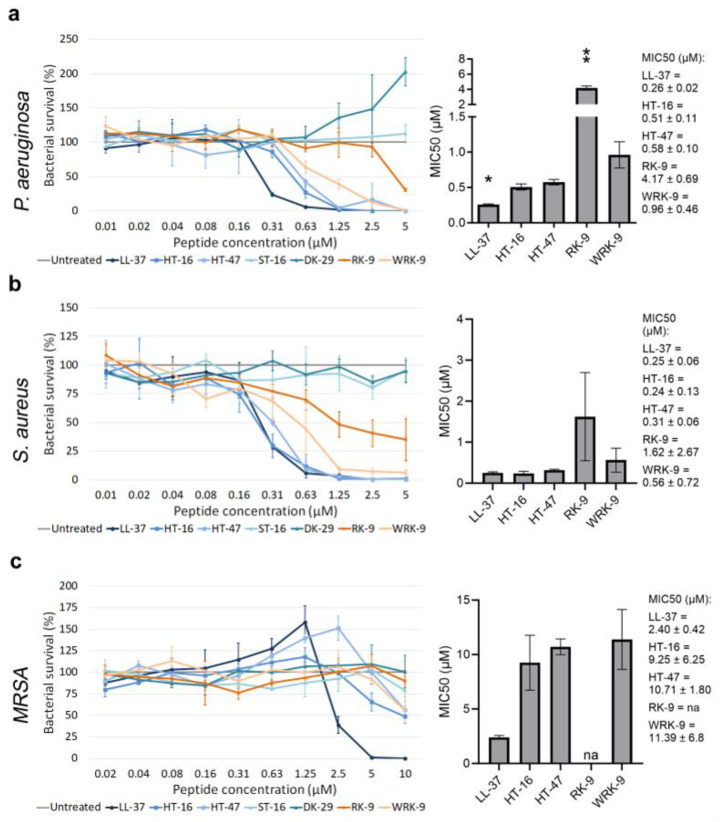
Determining the antimicrobial activity of modified CLEC3A-derived peptides RK-9 and WRK-9. (**a**) Antimicrobial activity assay using LL-37 (* *p* = 0.028), HT-16, HT-47, ST-16, DK-29, RK-9 (** *p* = 0.005), and WRK-9 in *P. aeruginosa*. The bar chart shows the determined MIC50 of the peptides. (**b**) Antimicrobial activity assay using LL-37, HT-16, HT-47, ST-16, DK-29, RK-9, and WRK-9 in *S. aureus*. The bar chart shows the determined MIC50 of the peptides (**c**) Antimicrobial activity assay using LL-37, HT-16, HT-47, ST-16, DK-29, RK-9, and WRK-9 in *MRSA*. The bar chart shows the determined MIC50 of the peptides including the numerical values with confidence intervals as the legend. Depicted are averages and standard deviations. Statistical significance was calculated using Prism 9 with a paired ANOVA test followed by a Dunett test and multiple comparisons, comparing each peptide with HT-16. All experiments were repeated three times (n = 3). na: not applicable.

**Figure 3 antibiotics-12-01532-f003:**
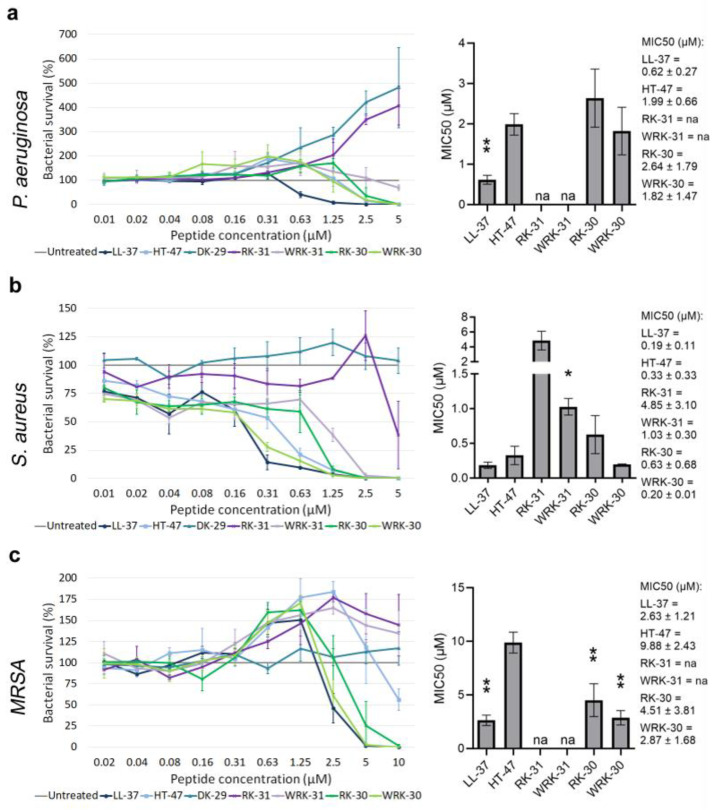
Determining the antimicrobial activity of modified CLEC3A-derived peptides RK-31, WRK-31, RK-30, and WRK-30. (**a**) Antimicrobial activity assay using LL-37 (** *p* = 0.009), HT-47, DK-29, RK-31, WRK-31, RK-30, and WRK-30 in *P. aeruginosa*. The bar chart shows the determined MIC50 of the peptides. (**b**) Antimicrobial activity assay using LL-37, HT-47, DK-29, RK-31, WRK-31 (* *p* = 0.042), RK-30, and WRK-30 in *S. aureus*. The bar chart shows the determined MIC50 of the peptides including the numerical values with confidence intervals as the legend. (**c**) Antimicrobial activity assay using LL-37 (** *p* = 0.003), HT-47, DK-29, RK-31, WRK-31, RK-30 (** *p* = 0.009), and WRK-30 (** *p* = 0.001) in *MRSA*. The bar chart shows the determined MIC50 of the peptides. Depicted are averages and standard deviations. Statistical significance was calculated using Prism 9 with a paired ANOVA test followed by a Dunett test and multiple comparisons, comparing each peptide with HT-47. All experiments were repeated three times (n = 3). na: not applicable.

**Figure 4 antibiotics-12-01532-f004:**
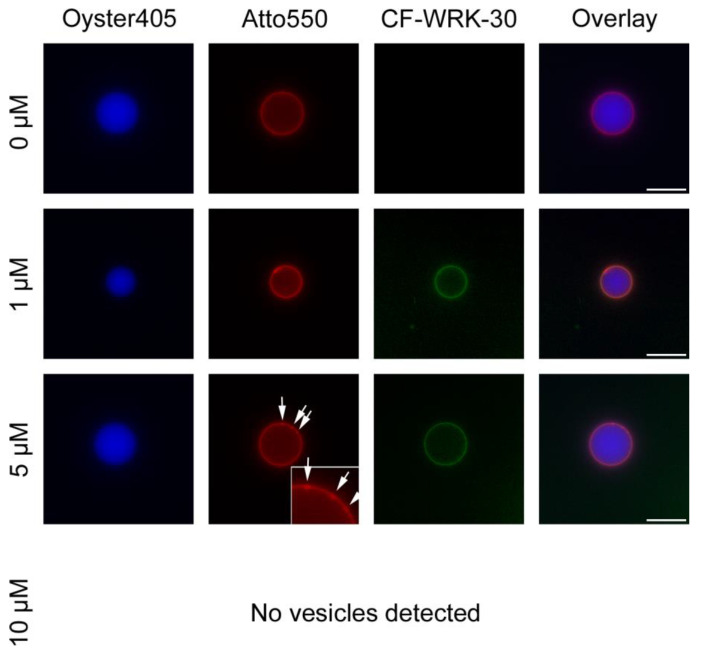
Interaction of WRK-30 with giant unilamellar vesicles. Giant unilamellar vesicles (GUVs) filled with dextran buffer containing Oyster405 (blue) and an anionic lipid membrane stained with Atto550 (red) were incubated for 30 min at room temperature with 0 µM, 1 µM 5 µM, and 10 µM of CF-labeled WRK-30 (green) and pictures were taken with a fluorescence microscope using a 60× magnification. White arrows and zoom-in mark sites of membrane lysis. Scale bars shown in the overlay correspond to 20 µm. The experiment was performed in three technical replicates (n = 3), except for 5 µM (n = 1).

**Figure 5 antibiotics-12-01532-f005:**
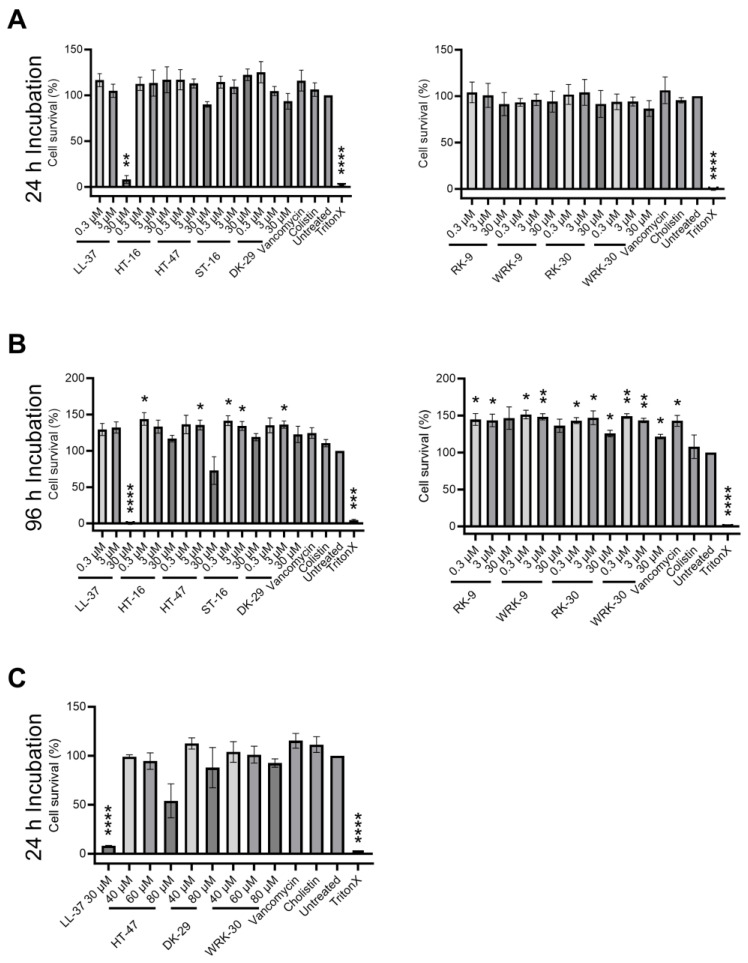
Cytotoxicity of modified CLEC3A-derived peptides. (**A**) Cytotoxicity assay of NIH3T3 cells with LL-37 (30 µM ** *p* = 0.003), HT-16, HT-47, ST-16, DK-29, RK-9, WRK-9, RK-30, WRK-30, Vancomycin, Colistin, Untreated, and TritonX (**** *p* < 0.0001). (**B**) Cytotoxicity assay of NIH3T3 cells LL-37 (30 µM **** *p* < 0.0001), HT-16 (0.3 µM * *p* = 0.050), HT-47 (3 µM * *p* = 0.044), ST-16 (0.3 µM * *p* = 0.032, 3 µM * *p* = 0.040), DK-29 (3 µM * *p* = 0.025), RK-9 (0.3 µM * *p* = 0.038, 3 µM * *p* = 0.045), WRK-9 (0.3 µM * *p* = 0.016, 3 µM ** *p* = 0.010), RK-30 (0.3 µM * *p* = 0.011, 3 µM * *p* = 0.046, 30 µM * *p* = 0.035), WRK-30 (0.3 µM ** *p* = 0.005, 3 µM ** *p* = 0.006, 30 µM * *p* = 0.024), Vancomycin (* *p* = 0.035), Cholistin, Untreated, and TritonX (*** *p* = 0.0002, **** *p* < 0.0001). (**C**) Cytotoxicity assay of NIH3T3 cells with LL-37 (**** *p* < 0.0001), HT-47, DK-29, and WRK-30, Vancomycin, Colistin, Untreated, and TritonX (**** *p* < 0.0001). All values are percentages normalized to the untreated controls (untreated controls were set as 100%). Depicted are averages and standard deviations. Statistical significance was calculated using Prism 9 with a paired ANOVA test followed by a Dunett test and multiple comparisons, comparing each treatment with the untreated control. All experiments were repeated three times (n = 3).

**Figure 6 antibiotics-12-01532-f006:**
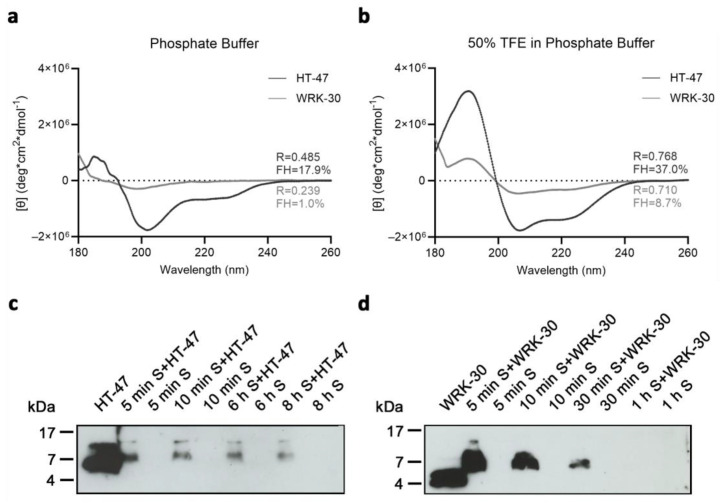
Circular dichroism (CD) spectroscopy of CLEC3A-derived peptides HT-47 and WRK-30 and their biostability in murine serum. (**a**) CD spectroscopy of HT-47 (black) and WRK-30 (dark grey) dissolved in phosphate buffer (PB). (**b**) CD spectroscopy of HT-47 (black) and WRK-30 (dark grey) dissolved in phosphate buffer with 50% TFE (50% TFE in PB). All CD spectroscopy experiments were performed three times (n = 3). (**c**) Representative immunoblot of HT-47 samples incubated in murine serum (S). Shown is an exposure time of 1 min. (**d**) Representative immunoblot of WRK-30 samples incubated in murine serum (S). Shown is an exposure time of 10 min. Images of immunoblots are cropped in width and length. Full-sized blots and secondary antibody-only controls are shown in Appendix A. Immunoblot experiments were performed three times (n = 3).

**Figure 7 antibiotics-12-01532-f007:**
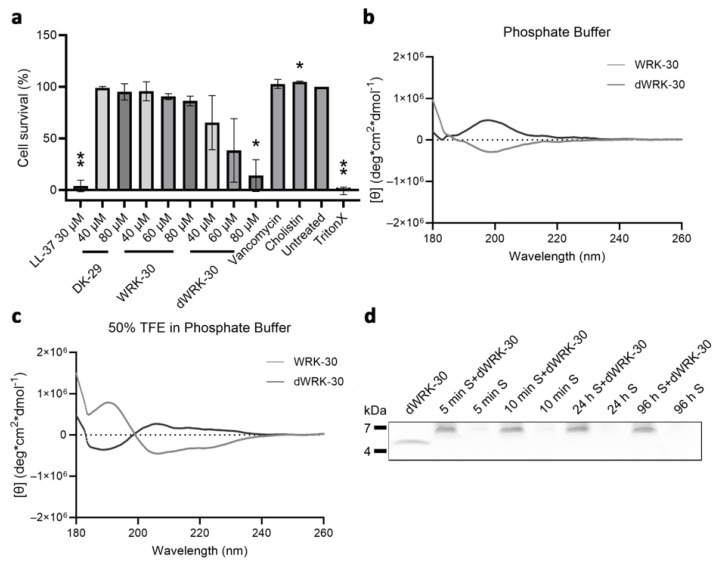
Cytotoxicity, CD spectroscopy, and biostability of modified CLEC3A-derived peptide dWRK-30. (**a**) Cytotoxicity assay of NIH3T3 cells with LL-37 (** *p* = 0.0038), DK-29, WRK-30, and dWRK-30 (* *p* = 0.0354), Vancomycin, Colistin (* *p* = 0.0219), Untreated, and TritonX (** *p* = 0.0015). All values are percentages normed to the untreated controls (untreated controls were set as 100%). Depicted are averages and standard deviations. Statistical significance was calculated using Prism 9 with a paired ANOVA test followed by a Dunett test and multiple comparisons, comparing each treatment with the untreated control. All experiments were repeated three times (n = 3). (**b**) CD spectroscopy of dWRK-30 (black) and WRK-30 (dark grey) in phosphate buffer (PB). (**c**) CD spectroscopy of dWRK-30 (black) and WRK-30 (dark grey) in 50% TFE in PB. All CD spectroscopy experiments were performed three times (n = 3). (**d**) Representative SDS-PAGE of dWRK-30 samples incubated in murine serum (S) The image of the SDS-PAGE is cropped in width and length. Full-sized SDS-PAGE gels are shown in Appendix A. SDS-PAGE experiments were performed three times (n = 3).

**Figure 8 antibiotics-12-01532-f008:**
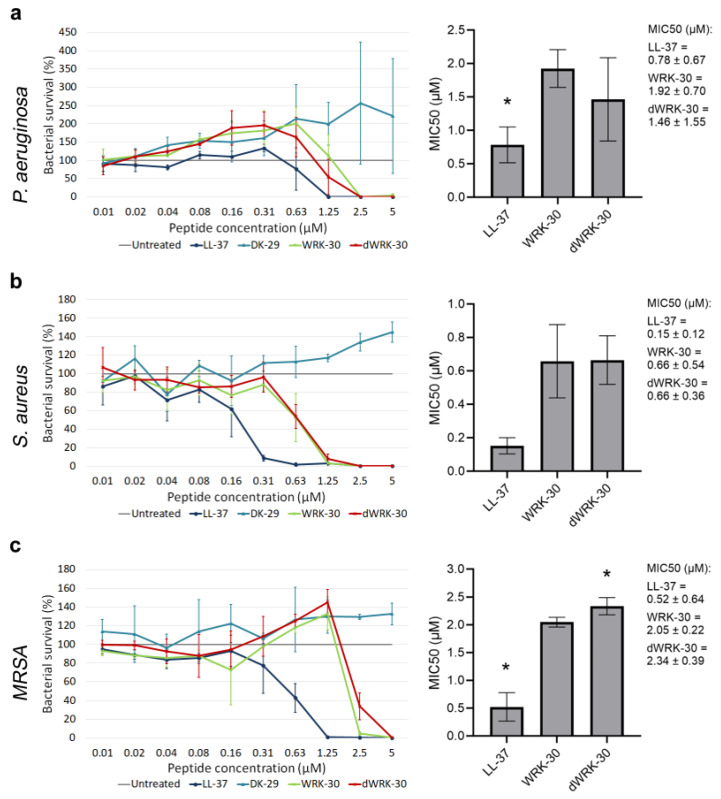
Determining the antimicrobial activity of modified CLEC3A-derived peptide dWRK-30. (**a**) Antimicrobial activity assay using LL-37 (* *p* = 0.020), DK-29, WRK-30, and dWRK-30 in *P. aeruginosa*. The bar chart shows the determined MIC50 of the peptides. (**b**) Antimicrobial activity assay using LL-37, DK-29, WRK-30, and dWRK-30 in *S. aureus*. The bar chart shows the determined MIC50 of the peptides. (**c**) Antimicrobial activity assay using LL-37 (* *p* = 0.024), DK-29, WRK-30, and dWRK-30 (* *p* = 0.027) in *MRSA*. The bar chart shows the determined MIC50 of the peptides including the numerical values with confidence intervals as legend. Depicted are averages and standard deviations. Statistical significance was calculated using Prism 9 with a paired ANOVA test followed by a Dunett test and multiple comparisons, comparing each peptide with WRK-30. All experiments were repeated three times (n = 3).

**Table 1 antibiotics-12-01532-t001:** Overview of native and modified CLEC3A-derived peptides. A comparison of the amino acid sequences of CLEC3A-derived peptides (ST-16, HT-16, DK-29, and HT-47) and modified CLEC3A-derived peptides (RK-9, WRK-9, RK-31, WRK-31, RK-30, and WRK-30). All sequences are shown from N- to C-terminus. Average molecular mass (MM) in Dalton (Da) was measured by LC–MS and the net charge was calculated using the peptide property calculator from Innovagen (http://www.pepcalc.com/, accessed on 30 April 2023).

Peptide	Amino Acid Sequence	MM (Da)	Net Charge pH = 7.0
**HT-16**	HTSRLKARKHSKRRVR	2016.6	+8
**RK-9**	RKHSKRRVR	1222.6	+6
**WRK-9**	WRKHSKRRVR	1408.8	+6
**HT-47**	HTSRLKARKHSKRRVRDKDGDLKTQIEKLWTEVNALKEIQALQTVCL	5544.1	+6
**RK-31**	RKHSKRRVRDKDGDLKTQIEKLWTEVNALKEI	3891.9	+4
**WRK-31**	WRKHSKRRVRDKDGDLKTQIEKLWTEVNALKEI	4078.1	+4
**RK-30**	RKHSKRRVR-GGG-LKTQIEKLWTEVNALKEI	3532.6	+6
**WRK-30**	WRKHSKRRVR-GGG-LKTQIEKLWTEVNALKEI	3718.8	+6
**ST-16**	STSQLQAQQSSQQQVQ	1775.8	±0
**DK-29**	DKDGDLKTQIEKLWTEVNALKEIQALQTVCL	3544.0	−2

## Data Availability

All data generated or analyzed during this study are included in this published article (and its Appendix A files).

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
