# Peer review of "Modified CLEC3A-Derived Antimicrobial Peptides Lead to Enhanced Antimicrobial Activity against Drug-Resistant Bacteria"

_antibiotics, 2023, doi:10.3390/antibiotics12101532_

Round 1

Reviewer 1 Report

The aim of this work is to find novel peptides with improved antimicrobial and therapeutical properties, by making modifications of already known antibiotic peptides. The rational and design of the project is good, including the peptide design, the antimicriobial and cytotoxic assays, as well as the structural studies by CD.

Although the authors do not achieve to improve the parent peptides in all of its therapeutic properties, they identified a candidate (WRK-30) with better antimcrobial potency and lower cytotoxicity. Unfortunately, this peptide shows less serum stability than the parent peptide, but it paves the way to explore new modifications that allows to keep the therapeutic properties gaining more stability.

I would like to make some comments, line by line:

-Line 13: Where you say "biological properties" I think you should say "therapeutic properties" since you are evaluating MIC50, cytotoxicity and serum stability, which are mainly therapeutic features, more than biological. This suggestion is applies to the same expression found in lines 22, 61, 65, 328

-Line 20: Scientific species names should be written in italics, in this case, S. aureus. This applies also to lines 119, 135, 139, 157, 427, 428 

-Line 21: The expression "in vivo" should be written in italics. Also in line 403, 406, 424.

-Line 21: "We identify" should be "We identified"

-Line 37: Respect to the statement "AMPs act by membrane destabilization and pore formation": The first part is true, there is always membrane destabilization when an AMP interacts with the membrane, but the second part it is not. Many AMPs form pores but many others not, especially short peptides. There are other well-known mechanisms of action described in the literature. You cand find a summary of them in: 

* Shai, Y. (2002). Mode of action of membrane active antimicrobial peptides. Peptide Science: Original Research on Biomolecules, 66(4), 236-248. doi: https://doi.org/10.1002/bip.10260

-Line 59: LL-37 peptides is mentioned twice in the same sentence, I suggest to avoid the repetition changing the sentence to: "the cytotoxic properties of LL-37 on eukaryotic cells limit the use of this peptide for therapeutic applications".

-Line 60: The word "common" appears wrongly hyphenated, "com-mon".

-Line 64: I suggest to change the beggining of the sentence from "We here apply" to "Here we apply".

-Line 70: In the section title the words "Antimicrobial Peptide" appear with the first letter in uppercase. There is no reason to write it like that, they should be lowercase. This also applies to line 114, 160, 183, 221, 262, 272, 426, 432, 442, 465, 481, 500, 520, 534, 550, 

-Figure 1: In the top panel of the figure, I suggest to put the protein structure at the topmost position and below the title "CLEC3A-derived peptides" followed by the scheme of the peptides. In addition, please indicate in the figure (not only in the caption) that the top scheme represent the whole protein. Please, add the meaning of the abbreviations contained in the protein scheme. And finally, please check the color code because you mention a dotted black box which is actually a dashed black box; then, you mention grey stripped areas that are not visible in the figure. Please make it more clear.

-Table 1: in the title of the rightmost column you should introduce a blank space between "Net" and "charge".

-Line 115: you should change "CLEC3A derived peptides" by "CLEC3A-derived peptides".

-Line 120: The word "MIC50" appears sometimes with number 50 as a subindex and sometimes, normal, please unify the notation.

-Figure 2: The label "a" of the top panel appears partially cropped. Please, abbreviate Pseudomonas aeruginosa as "P. aeruginosa", not "Ps. aeruginosa". Regarding the represented data, you are using LL-37 as a positive control and untreated bacteria, and peptides ST-16 and DK-29, as negative controls. I think LL-37 and untreated bacteria are good controls, but ST-16 and DK-29 are not because the reader doesn't know about their activity. I suggest to represent in the plots both controls: not only LL-37, but also untreated bacteria (even it is a straight line at 100%). And I suggest to change the name of control 1 and control 2, and rename with their real names, ST-16 and DK-29, and discuss later the observations on these peptides. On the other hand, in the bar plots representing MIC50, I recommend to include the confidence interval for MIC50 values (in the format MIC50 = XX ±XX uM), because it will help to understand the differences between peptides. I don't understand why LL-37 is statistically different in the first plot but not in the third one, where its MIC50 its clearly far from those from the other peptides. I would like to get a more detailed explanation of how are you calculating the statistical significance (I guess you compare every peptide in this figure to HT-16, as it is the parent peptide). In addition, please, when you give p values, round it up to only use a significative number (0.009, not 0.0094), because the numbers at the right of that number don't give any statistical information)

-Line 125: Abbreviation should be "P. aeruginosa" not "Ps. aeruginosa". This applies also for line 135, 139, 148, 157

-Figure 3: Abbreviation should be "P. aeruginosa" not "Ps. aeruginosa" and the same suggestions given for figure 2.

-Line 161: It should say "lipid-peptide", not "lipid peptide".

-Line 162: The word "composed" appears wrongly hyphenated as "com-posed".

-Figure 4: In the row corresponding to 5 uM, in the red channel there are two arrows pointing to regions with signs of membrane lysis, but I am note able to see those signs. It would be great to includes images with more evident signs of lysis.

-Supplementary figure 4: The scale of the y-axis should be adjusted, since the maximum of the curves is less than 1e-6, whereas the y-axis spans to 4e-6. Please, improve the format of the labels, changing expressions as x^x for superindex. The label of the y-axes indicates θ (which is ellipticity), but the units correspond to molar ellipticity which is symbolized as [θ], with brackets. Please correct this in all the figures with CD spectra. Also check the units, because traditionally molar ellipticity units are deg * cm^2 * dmol^-1, and you write deg * nm^2 * dmol^-1.

-Line 237: You say that the helices are weaker. I think this is not well expressed. CD results points to the formation of helical structure but the low intensities should be associated to low populations of structured peptides, not to "weak helices". So, you can say that HT-47 shows a higher population of structured peptides (forming helices) than RK-31, RK-30 and WKR-31, whose CD spectra point to a low population of structured peptides.

-Supplementary figure 5: Please improve the y-axis labelling avoiding the x^x format and using superindexes. I recommend to adjust the y-axis scale.

-Figure 5: This figure has a too long caption (it also applies to other figures).I think you include too many details in the caption that can be moved to materials and methods, or referenced to the material and methods section.

-Lines 263-270: in the immunoblots of figure 6c and 6d, we can see the first lane showing the corresponding native peptides HT-47 and WRK-30. In the sucessive lanes, the peptides are put in the presence of murine serum and the bands are visible until they dissapear, but in both cases it seems like the bands corresponding to the peptide have migrated less than the band of the control peptide in the absence of serum. What is your explanation for this? Maybe you can include it in the text.

-Figure 8a: Where you say "Tritron" I guess it is "Triton".

-Figure 8b: I suggest to superimpose the spectra of WRK-30 and dWRK-30 in the same plot to facilitate the visualization of the similarity.

-Figure 8c: the same as observed before: the peptide in the presence of serum displays bands that migrate less than the peptide in the absence of serum, what is your explanation for this?

-Supplementary figure 7 a: Why there is a band which should correspond to HT-47 in the lane labeled as 10 min S. The same for WRK-30. The band of the peptide should appear in the first and second lanes, but not in the third one. Why is this happening? In addition, in general, all the gels show a not very optimal aspect, you should have found the way to get better gels.

-Line 364: I don't think it is necessary to give the code for the amino acids, the audience of this article knows this code perfectly. Furthermore, you included a lot of peptide sequences before in the text and didn't explain the one-letter code for the amino acids. 

-Line 387: You say that the location of arginine residues in the plane of polar headgroups of the lipid bilayer favours the formation of cation-pi interactions, but with which residues? There is no pi systems in the peptide, only the tryptophan which is inserted in the bilayer.

-Line 412: CD-spectroscopy should not be hyphenated.

-Line 419: I agree with the fact that WRK-30 is the most interesting variant of HT-47, as it shows improved antimicrobial activity and less cytotoxicity, but please you should remark that it also shows worse serum stability than HT-47. This means that you were not able to make a full improvement of the therapeutic index of HT-47, but in the future, further modifications of WRK-30 could be explored to gain stability without losing antimicriobial efficienty and without becomeing more cytotoxic.

-Line 503: I think 10x10^3 is better expressed as 1x10^4.

-Line 517: you calculated the therapeutic index (TI), but I think you don't include TI values anywhere.

-Line 526: You say that CD data are shown in mean residue ellipticity (which should be symbolized as [θ]_MR or directly MRE), but the equation you indicate is the equation for the molar ellipticity. The mean residue ellipticity is normalized by the number of amino acids and it is: MRE = ((M / (N - 1)) * θ) / (10 * c * d), being M the molecular weight of the peptide and N the number of amino acids. Secondly, the units are right as you show them: deg * cm^2 * dmol^-1, so please correct your plots in which always write nm^2 instead of cm^2.

-Line 553: The word "obtained" appears wrongly hyphenated as "ob-tained".

-References: please, check the reference format because the journal name generally appears abbreviated, but sometimes not. 

Quality of English is good, perhaps some minor issues.

Reviewer 2 Report

1. General comment

In the manuscript, synthetic AMPs, CLEC3A derived AMPs HT-16 and HT-47, were prepared, and their antimicrobial activity against gram-positive and gram-negative bacteria, the cytotoxicity on murine cells, and the biostability of the modified peptides in serum were investigated. A novel antimicrobial peptide, WRK-30 showed enhanced antimicrobial potency against S. aureus and MRSA and less cytotoxic to eukaryotic cells. The results suggest that WRK-30 has potential as a candidate for future in vivo experiments.

2. Major revision

1)             Table 1

As the authors use Da, molecular weight should be revised to molecular mass.

It is recommended to explain the molecular mass of native and modified CLEC3A-derived peptides as both theoretical or calculated (monoisotopic or average) molecular mass by using the peptide property calculator from Innovagen and measured or observed (monoisotopic or average) molecular mass by using LC-MS.  

1. General comment

In the manuscript, synthetic AMPs, CLEC3A derived AMPs HT-16 and HT-47 were prepared, and their antimicrobial activity against gram-positive and gram-negative bacteria, the cytotoxicity on murine cells, and the biostability of the modified peptides in serum were investigated. A novel antimicrobial peptide, WRK-30 showed enhanced antimicrobial potency against S. aureus and MRSA and less cytotoxic to eukaryotic cells. The results suggest that WRK-30 has potential as a candidate for future in vivo experiments.

2. Major revision

1)             Table 1

As the authors use Da, molecular weight should be revised to molecular mass.

It is recommended to explain the molecular mass of native and modified CLEC3A-derived peptides as both theoretical or calculated (monoisotopic or average) molecular mass by using the peptide property calculator from Innovagen and measured or observed (monoisotopic or average) molecular mass by using LC-MS.  

Reviewer 3 Report

The article entitled “Modified CLEC3A derived Antimicrobial Peptides Lead to Enhanced Antimicrobial Activity against Drug resistant Bacteria” describes the application of the different sequence modifications of the CLEC3A derived AMPs HT 16 and HT 47 in order to increase biological properties in addition to cytotoxicity, secondary structure, and biostability of the modified CLEC3A derived peptides. The manuscript would be of general interest to the researchers of this field, but the manuscript lacks some information that the author should consider and incorporate in the present form of the manuscript. Here are some concerns that need to be addressed in the present form of the manuscript. With current version, the manuscript does not meet the requirements to be published in your journal.

Some comments and corrections for authors:

1.     Overally, the manuscript has a lot of punctuation and grammatical errors and needs to be corrected (i.e., there must be comma before and in all mns when mention about over two parameters). Please run throughout the mns.

2.     The selection criteria of NIH3T3 cells (murine fibroblasts) should be clarified.

3.     Biostability Assay should be explained and enlarged in detail.

The article entitled “Modified CLEC3A derived Antimicrobial Peptides Lead to Enhanced Antimicrobial Activity against Drug resistant Bacteria” describes the application of the different sequence modifications of the CLEC3A derived AMPs HT 16 and HT 47 in order to increase biological properties in addition to cytotoxicity, secondary structure, and biostability of the modified CLEC3A derived peptides. The manuscript would be of general interest to the researchers of this field, but the manuscript lacks some information that the author should consider and incorporate in the present form of the manuscript. Here are some concerns that need to be addressed in the present form of the manuscript. With current version, the manuscript does not meet the requirements to be published in your journal.

Some comments and corrections for authors:

1.     Overally, the manuscript has a lot of punctuation and grammatical errors and needs to be corrected (i.e., there must be comma before and in all mns when mention about over two parameters). Please run throughout the mns.

2.     The selection criteria of NIH3T3 cells (murine fibroblasts) should be clarified.

3.     Biostability Assay should be explained and enlarged in detail.

Reviewer 4 Report

Your research paper on modifying CLEC3A-derived antimicrobial peptides is well-structured and provides valuable insights into developing novel AMPs with improved properties. Here are some suggestions for further improvement:

Comments:

1. Cite recent literature (2021-2022). 

2. Include a clear hypothesis or research question in the introduction section.

3. Revise conclusion. 
